# Integrating questionnaire measures for transdiagnostic psychiatric phenotyping using *word2vec*

**Aaron Sonabend W.**[1], **Amelia M. Pellegrini**[2], **Stephanie Chan**[1], **Hannah E. Brown**[2,3,4], **James N. Rosenquist**[2,3], **Pieter J. Vuijk**[5], **Alysa E. Doyle**[5], **Roy H. Perlis**[2,3]*, **Tianxi Cai**[1,6]*

1 Department of Biostatistics, Harvard T.H. Chan School of Public Health, Boston, Massachusetts, United States of America, 2 Center for Quantitative Health, Massachusetts General Hospital, Boston, Massachusetts, United States of America, 3 Department of Psychiatry, Massachusetts General Hospital and Harvard Medical School, Boston, Massachusetts, United States of America, 4 Department of Psychiatry, Boston Medical Center, Boston University School of Medicine, Boston, Massachusetts, United States of America, 5 Center for Genomic Medicine, Massachusetts General Hospital, Boston, Massachusetts, United States of America, 6 Department of Biomedical Informatics, Harvard Medical School, Boston, Massachusetts, United States of America

* rperlis@partners.org (RHP); tcai@hsph.harvard.edu (TC)

**Data Availability Statement:** All relevant data are within the manuscript and its Supporting Information files.

## Abstract

### Background

Recent initiatives in psychiatry emphasize the utility of characterizing psychiatric symptoms in a multidimensional manner. However, strategies for applying standard self-report scales for multiaxial assessment have not been well-studied, particularly where the aim is to support both categorical and dimensional phenotypes.

### Methods

We propose a method for applying natural language processing to derive dimensional measures of psychiatric symptoms from questionnaire data. We utilized nine self-report symptom measures drawn from a large cellular biobanking study that enrolled individuals with mood and psychotic disorders, as well as healthy controls. To summarize questionnaire results we used word embeddings, a technique to represent words as numeric vectors preserving semantic and syntactic meaning. A low-dimensional approximation to the embedding space was used to derive the proposed succinct summary of symptom profiles. To validate our embedding-based disease profiles, these were compared to presence or absence of axis I diagnoses derived from structured clinical interview, and to objective neurocognitive testing.

### Results

Unsupervised and supervised classification to distinguish presence/absence of axis I disorders using survey-level embeddings remained discriminative, with area under the receiver operating characteristic curve up to 0.85, 95% confidence interval (CI) (0.74,0.91) using Gaussian mixture modeling, and cross-validated area under the receiver operating

**Funding:** This work was supported by a grant from the National Institutes of Mental Health (NIMH) and the National Human Genome Research Institute (NHGRI) (grant number: 1P50MH106933-01) to Dr. Perlis.

**Competing interests:** Dr. Perlis has served on advisory boards or provided consulting to Genomind, RID Ventures, and Takeda, and holds equity in Psy Therapeutics and Outermost Therapeutics. Dr. Perlis is an Associate Editor at JAMA Network Open. The other authors report no conflict of interest. This does not alter our adherence to PLOS ONE policies on sharing data and materials.

characteristic curve 0.91, 95% CI (0.88,0.94) using logistic regression. Derived symptom measures and estimated Research Domain Criteria scores also associated significantly with performance on neurocognitive tests.

## Conclusions

Our results support the potential utility of deriving dimensional phenotypic measures in psychiatric illness through the use of word embeddings, while illustrating the challenges in identifying truly orthogonal dimensions.

## Introduction

While much of medicine focuses on defining categories of illness, it is increasingly recognized that many aspects of psychopathology lie on a continuum with normal functioning, with such pathology best understood dimensionally[1]. This shift acknowledges the complexity of psychiatric nosology, where symptoms may not clearly distinguish specific disorders and may vary substantially within a single disorder[2–4].

The National Institutes of Mental Health (NIMH), through its Research Domain Criteria (RDoC) workgroups, originally enumerated five domains of neuropsychiatric functioning, suggesting assessments that measure aspects of each domain[5,6]. However, achieving a truly multidimensional comprehensive assessment would require so many measures as to be potentially intractable. Moreover, until domain-specific measures are developed and validated, most assessment tools are unlikely to correlate perfectly with the construct that a domain is intending to measure.

As part of a large cellular biobanking initiative, we developed a clinical assessment battery based on self-report measures intended to sample aspects of each domain. In the present study, we developed a novel method for succinctly *summarizing* an individual's symptom profile based on self-report questionnaires that leverages word embeddings derived from natural language used in medical and psychiatric literature. Word embeddings consist of a numerical representation of words as vectors in a low-dimensional space. This technique aims to capture the distributional semantic relationship of language[7]. That is, the semantic and syntactic relationship between words is preserved in this latent space; word similarity is reflected as distance, facilitating the detection of word meaning and contextual connotation. Additionally, vector operations are equivalent to word associations. For example, simple analogies such as, "Paris is to France as Madrid is to *Spain*" and "man is to king as woman is to *queen*", can be easily answered through embedding vector operations.

Based on word-embeddings, we derive multiaxial scores to represent these symptom severity profiles from self-report questionnaires. Our embedding-based approach enables us to leverage similarities and relatedness of different questions when summarizing a patient profile. This should enable extraction of different components of the questions to reflect specific aspects of the questionnaires such as correspondence to RDoC domain, which is not feasible using raw questionnaire scores alone. We validated the clinical utility of these embeddings scores by evaluating the association between the scores and standard categorical diagnoses, both in terms of lifetime and current illness. Further, we examined the relationship between our embedding-based measures and the derived RDoC scores with an objective neurocognitive testing battery in order to investigate the extent to which cognition might associate with symptoms in multiple domains.

## Methods and materials

### Study design and cohort generation

We enrolled 310 participants between 2009 and 2018 from outpatient psychiatric clinics of the Massachusetts General Hospital, as part of a broader clinical assessment for a cellular biobanking study, and via advertisements seeking healthy control (HC) participants[8]. All participants signed written informed consent and the study protocol was approved by the Partners HealthCare Institutional Review Board (protocol #: 2009P000238).

Participants were age 18–70 years and drawn from the following lifetime diagnostic categories: major depressive disorder (n = 68), bipolar I or II disorder (n = 55), schizophrenia or schizoaffective disorder (n = 75), as well as HCs with none of these (n = 112).

### Assessments

Categorical diagnosis (or lack of diagnosis for HCs) was confirmed by a psychiatrist rater, using the Structured Clinical Interview for DSM-IV (SCID) and the MINI Neuropsychiatric Interview (MINI) Version 5.0.0[9,10]. Participants were designated as HCs if they had no psychiatric diagnosis based on the SCID or MINI, and no intellectual disability (defined by Full-Scale Intelligence Quotient <70)[11].

For all participants, nine assessments of symptom severity were administered: the Berkeley Expressivity Questionnaire (BEQ), a 16-item scale designed to measure an individual's emotional expressivity[12]; the Behavioral Inhibition System and the Behavioral Approach System (BISBAS), used to measure two general motivational systems of behavior[13]; the Depression Anxiety and Stress Scale (DASS), a 42-item questionnaire used to measure the negative emotional states of depression, anxiety, and stress[14]; the Obsessive-Compulsive Inventory—Revised (OCIR), an assessment of common obsessive-compulsive disorder (OCD) symptoms (i.e., washing, checking, ordering, obsessing, hoarding, and neutralizing)[15]; Patient-Reported Outcomes Measurement Information System (PROMIS)–Anger, a measurement of angry mood (e.g., irritability, frustration), negative social cognitions (e.g., interpersonal sensitivity, envy, disagreeableness), and efforts to control anger[16]; PROMIS—Applied Cognition, a measurement of self-perceived functional abilities with regard to cognitive tasks (e.g., memory, concentration)[16]; PROMIS—Social Isolation, a measurement of perceptions of being avoided, excluded, disconnected from, or unknown by others[16]; the Cohen Perceived Stress Scale (PSS), a 10-item measurement of the degree to which situations in one's life are considered stressful[17]; and the Social Responsiveness Scale—Second Edition (SRS2), used to distinguish autism spectrum conditions from other psychiatric conditions[18].

All participants also completed a battery of measures drawn from the Cambridge Neuropsychological Test Automated Battery (CANTAB), a well-validated set of paradigms spanning multiple domains of cognition relevant to healthy individuals and those with psychiatric disorders[19–21]. These included Reaction Time (RTI), Spatial Working Memory (SWM), Stop Signal Task (SST), Rapid Visual Information Processing (RVP), Paired Associates Learning (PAL), Intra-Extra Dimensional Set Shift (IED), Attention Switching Task (AST), Emotion Recognition Task (ERT), and Affective Go/No-go (AGN).

### Natural language processing

Word embeddings (500 dimensions) were trained via the Word2Vec algorithm[22,23] using approximately 500,000 biomedical journal articles published by Springer between 2006 and 2008 (Springer Nature Switzerland AG.), and the introduction to the DSM-IV[24]. The entire corpus of text was first pre-processed by removing all punctuation, then lexical variant

generator (LVG) lexical tools[25] were used to normalize the words to unigrams. We applied the skip-gram neural network model in the Word2Vec package to the normalized text to train embedding vectors for words with a minimum total frequency of 100 in the text. Our model used a window of size 10 and negative sampling with 10 noise words sampled. We iterated over the entire corpus 5 times[26,27].

The resulting embeddings were used to transform the above-mentioned nine questionnaires into the embedding space, which was then used for assessing psychiatric disorders. Question-level vectors were generated within each survey as the weighted sum of the tokenized word vectors in the question with weights being the inverse document frequency (IDF)[28] computed in the original Springer training corpus. For the analysis, we only considered questions that had ordinal response type. Survey-level embeddings were computed for each patient as the weighted sum of the question vectors multiplied by an integer reflecting their ordinal answer (Fig 1). Thus, the outcome for each patient's survey response is summarized in a 500-dimensional embedding vector. The end result is represented in a matrix for each survey, containing information for all patients. Each questionnaire matrix has dimension $310 \times 500$, which corresponds to the number of patients by the embedding dimension.

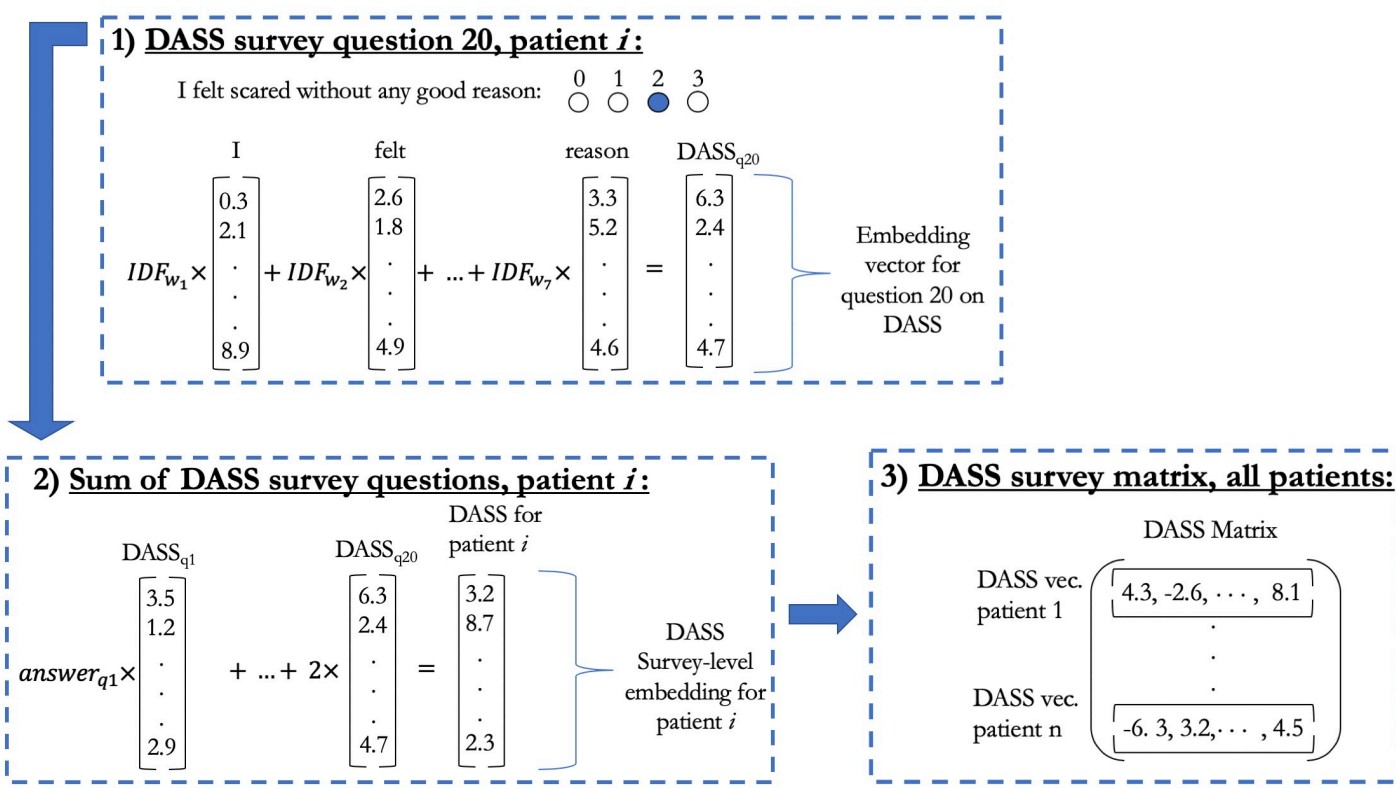

**Fig 1. Workflow to compute survey-level embedding matrix.** The example is illustrated with the DASS questionnaire and a given patient *i* response. In Box 1 we illustrate the first step with question 20: "I felt scared without any good reason." We weight the 500-dimensional embedding vector for each word in the question by its corresponding IDF weight, this yields a question-level embedding vector for question 20 (q20). Subsequently as shown in Box 2 we add all question vectors in the survey, respectively weighted by the patient's i answers. In this example, answer to q20 was "2," thus we multiply the last question embedding by 2. The result is a survey-level embedding vector for DASS for patient i. We repeat the process with DASS for all patients *i = 1, . . .,310*, concatenating the transposed survey-level vectors as shown in Box 3. Finally, we have a questionnaire-level matrix of dimension 310 x 500, corresponding to patient by embedding dimension.

## Analysis

Principal component (PC) analysis was performed on the embedding matrix of each survey for dimensionality reduction. PC analysis captures the maximum amount of variance contained in the matrix, however this is not necessarily information related to symptoms. To capture information in the questionnaires relevant to self-rated symptoms in our dimensionality reduction of the survey matrices, canonical correlation analysis was carried out for each questionnaire. For this we used the patient response matrix for the Diagnostic and Statistical Manual of Mental Disorders (DSM-V) Self-Rated Symptom Measure (13 DSM-V domains in total) and the leading embedding PCs[29]. The first canonical correlation variable (CCV) and the first 1, and 2 PCs of this same matrix were used for exploratory data analysis.

We also use our multidimensional symptom profiles for constructing scores associated with each RDoC category. We calculate the RDoC scores based on cosine similarity between a reference RDoC domain vector and the patient RDoC domain specific embedding vector. These RDoC domain embedding vectors are based on weighted averages of the embedding vectors for the seed words representative of the RDoC domains as given in[30], with IDF as weights. We proceeded to generate vectors for patients corresponding to each domain. These are defined as the weighted average of all question embeddings, with the weights as the product of the patient's ordinal response and the cosine similarity between the question and the domain embeddings.

## Clustering

Patients were grouped a priori into three diagnostic categories: supercontrols for individuals with no SCID or MINI current or lifetime axis I diagnoses; controls for subjects with no current or lifetime diagnosis of major depressive disorder, bipolar disorder, or schizophrenia or schizoaffective disorder, but with at least one lifetime MINI axis I diagnosis (e.g., substance use disorder, panic disorder); and cases for subjects diagnosed with major depressive disorder, bipolar disorder, or schizophrenia or schizoaffective disorder. Unsupervised clustering was carried out using the following types of features: the first CCV for each survey, and the first $n = 1$, 2 PCs for each survey matrix, rendering a 9- and 18-dimensional feature-space, respectively. Gaussian mixture model (GMM) clustering was fit assuming two clusters. This process is repeated 500 times using bootstrapped samples.

We considered two sets of contrasts: cases vs. controls and supercontrols (C vs. CS), and controls and cases vs. supercontrols (CC vs. S). Performance in accuracy, area under the receiver operating characteristic curve (AUC), false positive rate (FPR) and false negative rate (FNR), bootstrap mean values and 95% confidence intervals (CI) are reported. For accuracy, FPR and FNR we use class probability of 0.5 as the determining threshold. In other words, patients are assigned to the cluster to which they most likely belong according to the model. To visualize how well the GMM clustering distinguishes groups, we further reduce the space spanned by the 18 PCs from questionnaires into two dimensions using t-Distributed Stochastic Neighbor Embedding (tSNE). Distribution of patients in this low-dimensional space is shown along with Gaussian contours from the GMM clustering.

Supervised classification using logistic regression was performed using the same contrasts described above over 500 bootstrap replications. The 10-fold cross validated accuracy, AUC, FPR and FNR results are shown for outcome prediction using four different sets of features for each survey; the first CCV, and the first 1, and 2 PCs. We report bootstrap mean and 95% confidence intervals.

### Neurocognitive CANTAB measures

For further validation, we then sought to estimate the association of individual scales, and their estimated RDoC domain scores, with individual CANTAB measures. Additionally, we explore the scales' aggregate association with neurocognitive scores. Association between the questionnaire embeddings, RDoC scores, and neurocognitive outcome measures from the CANTAB data is explored by computing the rank correlation between the neurocognitive outcome measures with both the first two PCs of the survey matrices and the RDoC category scores.

Combined questionnaire association to CANTAB measures is shown using a penalized linear regression. In particular, least absolute shrinkage and selection operator (LASSO) penalized linear regression was fit to each neurocognitive outcome measure. The leading two PCs from each of the nine questionnaires were used as features, and the outcomes with skewed distributions were log-transformed.

The LASSO regularization parameter for each regression was chosen to optimize the Akaike information criterion. The predictive power of the 18 PCs for each of the CANTAB measures was summarized based on the cross-validated rank correlation between the predicted and observed outcome; this was done over 500 bootstrap replications.

## Results

We first examined the ability of derived dimensional measures to retain discrimination between disease and non-disease states, using the raw ordinal responses as a benchmark for comparison. Table 1 shows the performance of GMM and logistic regression clustering using the first n = 1, 2, PCs from the nine questionnaires (F1), and the CCV (F2) as features in distinguishing CC vs. S and cases from C vs. CS. The embedding-based measures show better or similar classification performance across clustering methods. For supervised clustering, embedding-based measures do better than raw scores in both AUC and accuracy when classifying CC vs. S.

Supervised results from the 10-fold cross-validated logistic regression are shown in Table 1. Permutation tests were used to test the difference in accuracy and AUC bootstrap means between the embedding scores and raw response scores. Word embedding features showed classification power across the number of PCs as well as in the canonical correlation vectors. Maximum accuracy was obtained using 2 PCs per questionnaire: 0.86, 95% CI (0.81, 0.91) for both C vs. CS and CC vs S compared to clustering with raw ordinal scores which have accuracy of 0.85, 95% CI (0.81, 0.9) and 0.8, 95% CI (0.7, 0.88) for C vs. CS and CC vs. S, respectively. Both are significant differences based on the permutation tests (p-val<0.01). Highest AUC with embedding measures was obtained using the first CCV: 0.91, 95% CI (0.88, 0.94) and 0.89, 95% CI (0.85, 0.93) for C vs. CS and CC vs S, respectively. These scores are significantly higher (p-val<0.01) than classifying using raw questionnaire responses which yielded AUC of 0.75, 95% CI (0.69, 0.81) and 0.76, 95% CI (0.7, 0.82) for C vs. CS and CC vs S, respectively.

Low-dimensional embedding features still enabled discrimination between individuals with and without current or lifetime neuropsychiatric disorders. For example, the AUC for unsupervised classification using CCV, increased from 0.75, 95% CI (0.58, 0.93) and 0.76, 95% CI (0.59, 0.94) to 0.84, 95% CI (0.71 0.92) and 0.84, 95% CI (0.73 0.91) for C vs. CS, for CC vs. S (both differences have p-val<0.01). Unsupervised classification AUC using 2PCs for separating C vs. CS based on 1 PC was 0.84, 95% CI (0.72,0.92), compared to 0.83, 95% CI (0.69, 0.9) using ordinal scores (p-val < 0.01); there was no significant difference in AUC for discriminating between CC vs. S, both approaches yielded an AUC of 0.85, 95% CI (0.74, 0.91) and (0.73, 0.91) for embedding scores and raw scores respectively.

**Table 1. Accuracy (ACC), AUC, false positive rate (FPR), and false negative rate (FNR) are obtained using supervised (logistic regression) and unsupervised (Gaussian mixture modeling) classification.** All metrics are based on a probability of 0.5 classification threshold. Columns contain tables for clustering using the first canonical correlation variable and first n = 1, 2 principle components from the 9 questionnaires. Raw ordinal scores are used for comparison. Results are grouped into two column groups: the left group contains clustering comparing C vs. CS, the left group shows CC vs. S. Results for logistic regression are the mean of a 10-fold across-validation analysis.

| Supervised (logistic regression) | | | | | |
| --- | --- | --- | --- | --- | --- |
| C vs. CS | | | | | |
| CCV | | 1 PC | | 2 PCs | |
| Embeddings | Raw Scores | Embeddings | Raw Scores | Embeddings | Raw Scores |
| ACC 0.85 (0.8,0.89) | 0.73 (0.66,0.78) | 0.83 (0.78,0.87) | 0.83 (0.78,0.87) | 0.86 (0.81,0.91) | 0.85 (0.81,0.9) |
| AUC 0.91 (0.88,0.94) | 0.75 (0.69,0.81) | 0.89 (0.85,0.93) | 0.89 (0.85,0.93) | 0.9 (0.86,0.94) | 0.9 (0.86,0.94) |
| FPR 0.13 (0.08,0.19) | 0.16 (0.09,0.25) | 0.16 (0.1,0.22) | 0.15 (0.1,0.21) | 0.12 (0.07,0.18) | 0.13 (0.08,0.18) |
| FNR 0.18 (0.11,0.26) | 0.48 (0.34,0.65) | 0.19 (0.12,0.27) | 0.2 (0.12,0.29) | 0.17 (0.09,0.26) | 0.18 (0.11,0.27) |

| Supervised (logistic regression) | | | | | |
| --- | --- | --- | --- | --- | --- |
| CC vs. S | | | | | |
| CCV | | 1 PC | | 2 PCs | |
| Embeddings | Raw Scores | Embeddings | Raw Scores | Embeddings | Raw Scores |
| ACC 0.85 (0.8,0.89) | 0.77 (0.72,0.82) | 0.85 (0.8,0.89) | 0.8 (0.7,0.87) | 0.86 (0.81,0.91) | 0.8 (0.7,0.88) |
| AUC 0.89 (0.85,0.93) | 0.76 (0.7,0.82) | 0.89 (0.84,0.93) | 0.83 (0.78,0.88) | 0.89 (0.85,0.93) | 0.83 (0.78,0.88) |
| FPR 0.11 (0.07,0.15) | 0.08 (0.03,0.13) | 0.11 (0.07,0.15) | 0.04 (0,0.1) | 0.09 (0.05,0.13) | 0.05 (0,0.12) |
| FNR 0.29 (0.16,0.46) | 0.65 (0.46,0.85) | 0.28 (0.17,0.45) | 0.62 (0.27,1) | 0.26 (0.15,0.39) | 0.59 (0.22,1) |

| Unsupervised (Gaussian mixture model) | | | | | |
| --- | --- | --- | --- | --- | --- |
| C vs. CS | | | | | |
| CCV | | 1 PC | | 2 PCs | |
| Embeddings | Raw Scores | Embeddings | Raw Scores | Embeddings | Raw Scores |
| ACC 0.81 (0.75,0.86) | 0.7 (0.53,0.79) | 0.81 (0.74,0.86) | 0.79 (0.71,0.84) | 0.81 (0.75,0.86) | 0.77 (0.69,0.83) |
| AUC 0.84 (0.71,0.92) | 0.75 (0.58,0.93) | 0.84 (0.72,0.92) | 0.83 (0.69,0.9) | 0.83 (0.63,0.92) | 0.83 (0.74,0.89) |
| FPR 0.24 (0.11,0.36) | 0.33 (0.1,0.98) | 0.25 (0.14,0.4) | 0.27 (0.1,0.46) | 0.24 (0.1,0.47) | 0.2 (0.09,0.38) |
| FNR 0.16 (0.08,0.28) | 0.29 (0.1,0.63) | 0.16 (0.06,0.29) | 0.18 (0.06,0.360) | 0.15 (0.04,0.3) | 0.26 (0.09,0.39) |

| Unsupervised (Gaussian mixture model) | | | | | |
| --- | --- | --- | --- | --- | --- |
| C vs. CS | | | | | |
| CCV | | 1 PC | | 2 PCs | |
| Embeddings | Raw Scores | Embeddings | Raw Scores | Embeddings | Raw Scores |
| ACC 0.79 (0.72,0.85) | 0.7 (0.53,0.8) | 0.79 (0.71,0.86) | 0.79 (0.69,0.86) | 0.8 (0.71,0.86) | 0.76 (0.67,0.84) |
| AUC 0.84 (0.73,0.91) | 0.76 (0.59,0.94) | 0.85 (0.74,0.91) | 0.85 (0.73,0.91) | 0.83 (0.65,0.91) | 0.85 (0.78,0.9) |
| FPR 0.19 (0.06,0.32) | 0.3 (0.06,0.99) | 0.19 (0.07,0.34) | 0.19 (0.05,0.37) | 0.18 (0.04,0.42) | 0.11 (0.03,0.33) |
| FNR 0.22 (0.13,0.34) | 0.3 (0.12,0.55) | 0.21 (0.11,0.34) | 0.22 (0.1,0.4) | 0.21 (0.08,0.36) | 0.29 (0.12,0.42) |

Finally, for CCV, embedding based measures had 0.84, 95% CI (0.71,0.92) vs 0.75 95% CI (0.58, 0.93) for ordinal scores for separating C vs. CS (p-val<0.01). The GMM unsupervised clustering based on the two set of features, (F1) and (F2), yielded classifiers with very similar performances.

For illustration purposes, Fig 2 shows the distribution of CC vs. S with two GMM clusters on a two-dimensional space reduction of the first two PCs from the nine questionnaires. Gaussian contours are somewhat overlapping, and some cases behave as controls in this latent space. This shows both the hindrance in unsupervised clustering classification performance as well as the continuous severity measures; there is an evident distinction between the two clusters, even in this compressed space. Symptomatology as measured by the questionnaires appears distinct between groups and coincides largely with diagnostic category, especially for the cases, demonstrating information content in the continuous multiaxial measures (Table 1).

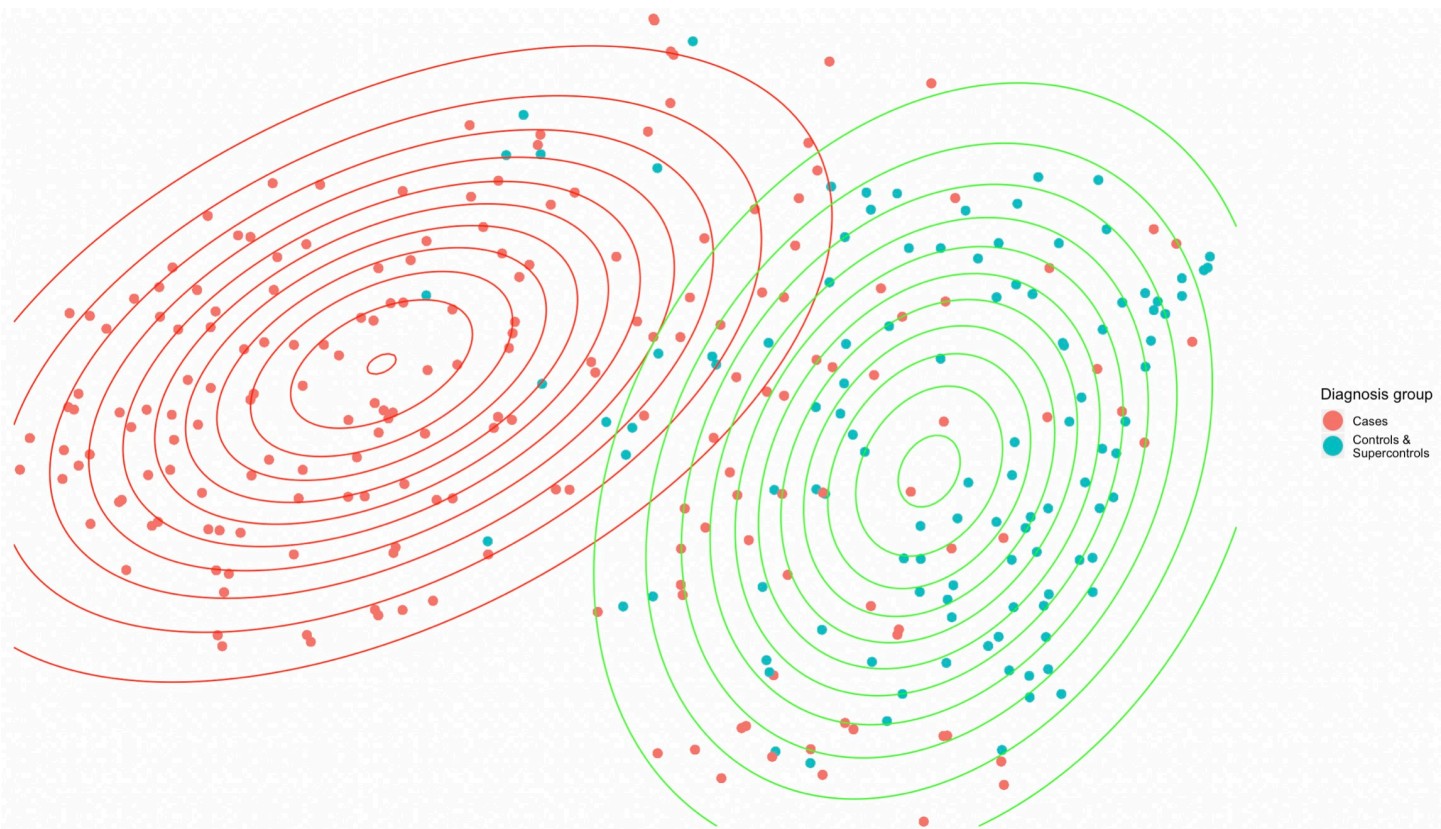

**Fig 2. t-Distributed Stochastic Neighbor Embedding is used for dimensionality reduction to map the 18 dimensions to 2, for illustrative purposes.** Contour colors represent the cluster assigned observations, point colors represent patient diagnosis groups.

Next, Fig 3 shows the rank correlation heatmap between the neurocognitive scores and each of the survey's first two PCs. We observed statistically significant correlation between tasks: RVP (both probability of hit and A'), PAL, IED, and ERT with questionnaire PCs: BEQ, PROMIS—Applied Cognition (PC1), and SRS2 (PC2). For example, RVP (both probability of hit [rho = 0.31, p < .001, rho = -0.42, p < .001] and A' [rho = 0.32, p < .001, rho = -0.42, p < .001]) are strongly correlated with PROMIS—Applied Cognition (PC1), and SRS2 (PC2), respectively.

We show the correlation between derived RDoC category scores and neurocognitive scores in Fig 4. We observed multiple statistically significant correlations between estimated RDoC scores and tasks. In particular, the Cognitive Systems correlated significantly with 11 of 14 tasks at the 0.1 level. Arousal Regulatory Systems, Positive Valence, and Systems for Social Processes correlated significantly at the 0.1 level with 6, 7 and 4 tasks, respectively. The lowest correspondence across cognitive tasks occurred in relation to the Negative Valence domain.

Finally, results in S2 Fig show to what extent the combination of the 18 PCs embedding features extracted from the nine questionnaires can jointly predict the neurocognitive scores and how each one contributes to the group of features. Fig 4 shows the rank correlations of predicted and observed CANTAB scores. There are seven CANTAB scores with out-of-sample rank correlation above 0.3. SWM (between errors), RVP (both probability of hit [0.46, 95% CI (0.34, 0.55)] and A' [0.48, CI (0.37, 0.57)]), PAL (total errors [0.33, CI (0.20, 0.45)] and 6 shapes [0.33, CI (0.20, 0.45)]), IED [0.38, CI (0.25, 0.49)], and ERT [0.49, CI (0.39, 0.59)] are relatively

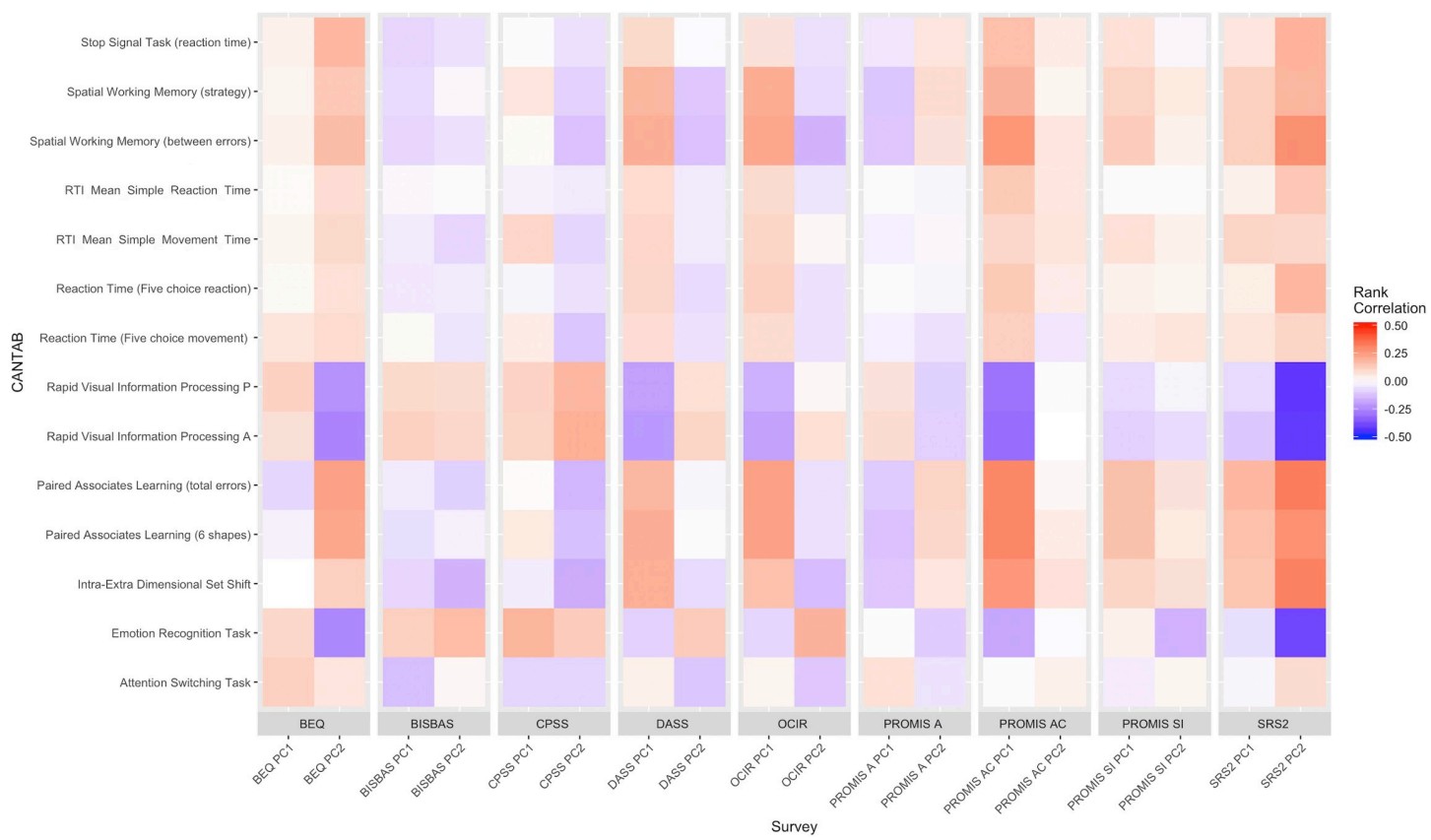

**Fig 3. Single survey rank correlation heatmap: We calculate the rank correlation between the first two principle components of the 9 questionnaires, and some selected CANTAB cognitive score metrics.** Positive correlations are indicated in red, negative correlations in blue, with strength of correlation indicated by color intensity.

well-explained by a linear combination of the embedding features. On the other hand, RTI with correlations below 0.2 with all bootstrap 95% CI containing 0.

Within the correlated measures, each questionnaire's PC captures an independent aspect that contributes to the prediction of a neurocognitive function. We confirmed this by using multivariate regression and analyzing the regression coefficients from each embedding feature, and by using the relative importance to how these jointly predict CANTAB measures. In particular, we also explored how the combination of embedding features are predictive of each CANTAB measure. Embedding features derived from five questionnaires are predictive of the IED task: BISBAS PC2 and OCIR PC2 have a negative association with a high neurocognitive score, whereas DASS, PROMIS—Social Isolation, and SRS2 are positively correlated. In the case of the OCIR, PC1 and PC2 are both independently associated with SWM (between errors). S1 Fig shows that the combination of embedding features is predictive of each CANTAB measure.

Well-predicted CANTAB measures, such as both RVP and ERT, were generally associated with a high number of predictive features. However, this was not always the case: for example, SWM was solely associated with the OCIR and SRS2. Thus, the observed high correlation is not always due to multiple, independent aspects measured by the questionnaires.

## Discussion

In this investigation of 310 individuals participating in a large cellular biobanking study, we developed an embedding-based tool to summarize potentially high-dimensional survey

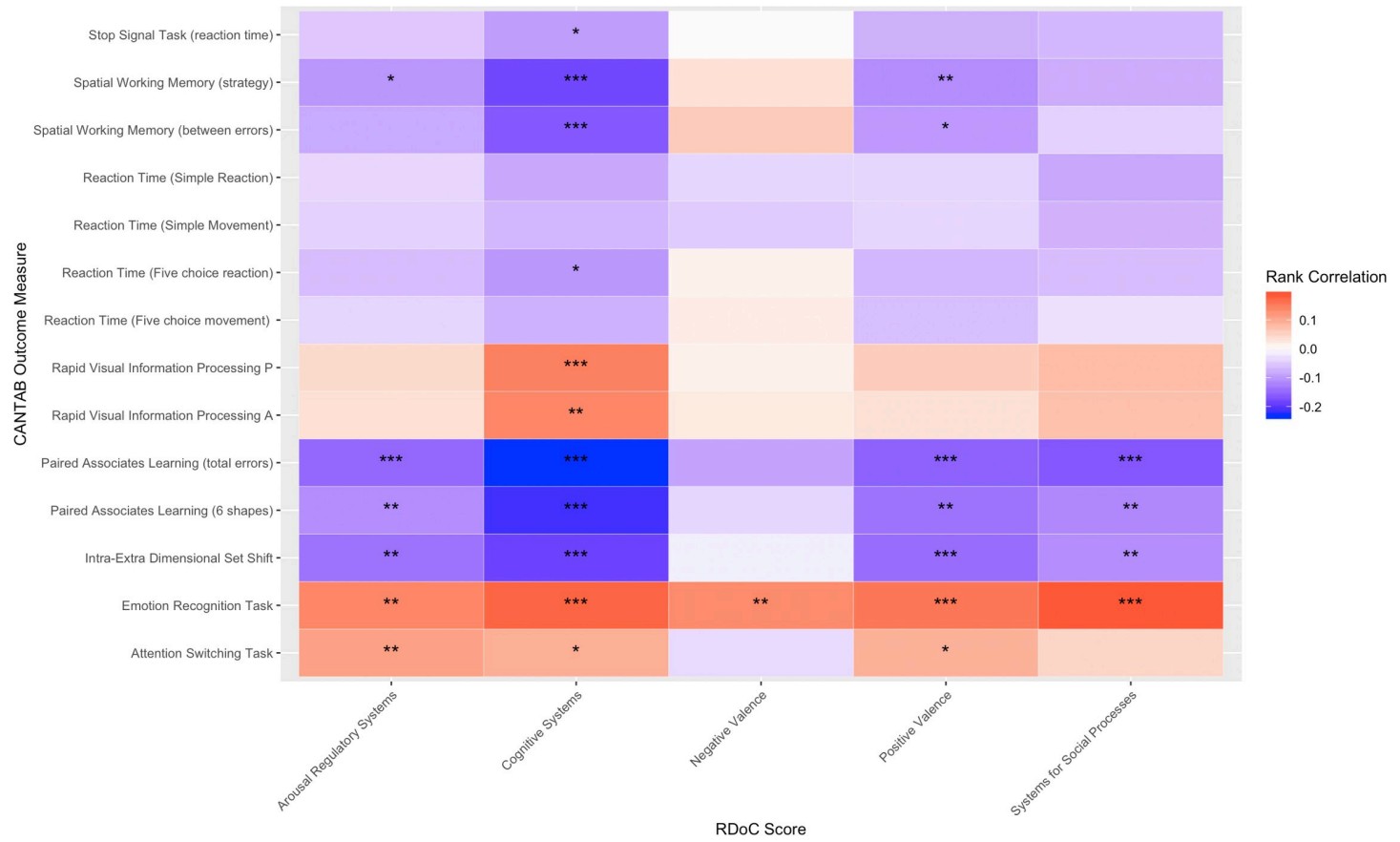

**Fig 4. RDoC score rank correlation heatmap: We calculate the rank correlation between the selected CANTAB cognitive score metrics and embedding-based derived RDoC scores.** High correlation is shown in warm colors. Significance is shown with stars: 0.1 \*, 0.05 \*\*, and 0.01 \*\*\*.

questionnaire data based on low-dimensional embedding scores. In line with the NIMH priorities emphasizing transdiagnostic application of Research Domain Criteria[31], our method provides dimensional rather than categorical approaches to disease. These scores can be used for downstream analyses as we showed with several use-cases, meant to illustrate the quantitative and qualitative advantages to integrating word embeddings to the questionnaire scores.

The embedding profiles generate transdiagnostic characteristics among individuals with current or lifetime psychiatric illness, as well as HC participants. As intended, our analyses suggest that the dimensional values still retain some ability to distinguish among individuals with current and lifetime psychiatric symptoms, enabling their application in both dimensional and categorical phenotyping studies. At the same time, consistent with the underlying motivation for applying the RDoC framework, these scores do not perfectly discriminate among diagnostic categories, nor distinguish individuals with an axis I diagnosis from HCs. That is, they carry additional information beyond categorical diagnosis.

Moreover, these symptom domains also correlate with neurocognitive measures, suggesting that the RDoC domains are not completely orthogonal. Moreover, similar to psychiatric diagnoses, which show overlapping areas of cognitive impairment[32], there is significant correlation between different symptom domains and objectively-measured tasks on the cognitive battery. In particular, we find that two domains focused on social functioning and expressivity, the BEQ and SRS, associate strongly with cognition, consistent with their relationship in

neurodevelopmental disorders such as schizophrenia or autism[33,34]. In a previous investigation including participants from this same study cohort, the association between CANTAB scores and performance on a delay-discounting task (a measurement of impulsivity) was examined transdiagnostically [29]. This prior work also demonstrated a correlation among cognitive domains that cut across traditional diagnostic categories, highlighting that domains of neuropsychological function are also not fully orthogonal.

With our method, we try to address two challenges that arise in survey scoring as it is currently performed: 1) questions which might be of different relevance towards contributing diagnostic information are equally weighted and counted for phenotyping, 2) discrete scores impose an artificial threshold for disease severity, leading to a coarse measure of patient behavior.

Regarding the difference in question relevance, a substantial benefit of incorporating embedding is to leverage semantic similarities between different words and contexts which is not possible with the ordinal scores. Thus, the information contained in each question is better captured. To understand what we refer to as semantic similarities in questions, consider the PROMIS A survey. Questions 3 and 5 both have a numerical response that range from 0 for "Never" to 4 for "Always;" they ask if in the past 7 days: "I felt like I was ready to explode" and "I felt annoyed," respectively. Qualitatively, these two questions use very different language. This is coded into the embedding space: the term "explode" is close to words like "annihilate," "destroy," and "vanish," while on the other hand "annoyed" is close to words like "frustrated," "irritated," and "displeased." Responding "often" to both of these questions might not mean the same in terms of anger severity. Embeddings provide a natural way to encode differences between these questions. The question meaning and importance for the diagnosis is learned from massive natural language sources, which is an improvement to either weighing them equally or imposing an artificial order. Other standard methods for summarizing questionnaires data, including principal components analysis, factor analysis and item response theory (IRT)[35–37], are effective in dimensionality reduction but do not leverage prior knowledge about the relatedness of the questions.

Furthermore, results shown in Table 1 suggest that incorporating embeddings into the classification generally improves the accuracy over the classification obtained based on dimension reduction of the ordinal scores with substantial improvements for some cases. The most substantial improvement is observed using the first canonical correlation variable (CCV) obtained from the scores and DSM-V Self-Rated Symptom Measure. Additionally, through our embedding profiles, we created estimated RDoC scores for individual patients based on their questionnaire data. We demonstrated the use of the RDoC scores by exploring their association with CANTAB scores. As expected, the Cognitive Systems domain is highly associated to neurocognitive performance.

Regarding the second challenge, our results also support the utility of continuous severity measures. In considering Fig 1; an illustrative example is the DASS questionnaire: in step 3 of the workflow we obtain a matrix with an embedding vector representing the status of each individual with respect to their depression and anxiety behavior. In other words, the survey embedding for a patient is the sum of question-level embeddings weighted by the response. In this survey-level embedding space, similar patients will cluster together, just like similar words do in the original space.

Likewise, difference in severity is illustrated in Fig 2, the GMM cluster distributions on the two-dimensional space reduction of the first two principle components from the nine questionnaires show how the two groups of patients behave differently. Most of the cases belong to the red cluster: in this reduced space it seems that there are few controls that behave as patients and many cases are both easily differentiable and well located within the red cluster. However,

there are cases that exhibit similar behavior as the controls which are located where the clusters intersect, and some cases are found entirely in the green cluster. This illustrates how aspects of psychopathology lie on a continuum with normal functioning, and psychopathology itself overlaps across traditional notions of disorder[1]. As mentioned previously, there is no prespecified threshold for illness and health. Additionally, the different symptom patterns between patients can be seen by the supervised and unsupervised clustering performance; difference in phenotypic behavior is well represented in the embedding space.

We note several limitations in the present work. First, while these participants were sampled from a large health system, they still represent a convenience sample such that the generalizability of our results in unselected populations remains to be established. Second, while questionnaires were selected in order to capture the breadth of RDoC domains, they do not represent complete assessments of any individual domain, as such a study (including all questionnaires and experimental paradigms for all domains) would likely be infeasible. Finally, we cannot exclude the possibility of treatment effects reflected in some domains, most notably the cognitive measures, as contributing to observed case-control differences. Follow-up work in larger cohorts will be valuable in characterizing such treatment effects.

Nonetheless, our results demonstrate the utility of applying an embedding approach, implemented in word2vec, to derive transdiagnostic measures from questionnaires in order to investigate RDoC dimensions. This method should enable efficient generation of new transdiagnostic measures drawing on already-validated scales, as well as derivation of such measures from existing data sets. More generally, our results support the relevance of considering clinical features that cross domains, by illustrating the limited ability of a diagnostic category alone to capture relevant clinical detail. Finally, they illustrate the complex relationship between domains, and particularly the correlation between symptom domains and cognition.

## Supporting information

**S1 Fig. Heatmap comparing coefficients from the regression described above using questionnaire principle component and canonical correlation variable features regressed on each of the CANTAB scores as outcomes.**
(TIFF)

**S2 Fig. Combined survey rank correlation: Correlation between real CANTAB scores and CANTAB scores predicted using first two principal components and first canonical correlation variable from each questionnaire.** Plot shows correlation over a 10-fold cross-validated regression for 500 bootstrapped samples. 95% bootstrap confidence intervals are shown in orange.
(TIFF)

## Author Contributions

**Writing – original draft:** Aaron Sonabend W., Roy H. Perlis, Tianxi Cai.

**Writing – review & editing:** Aaron Sonabend W., Amelia M. Pellegrini, Stephanie Chan, Hannah E. Brown, James N. Rosenquist, Pieter J. Vuijk, Alysa E. Doyle, Roy H. Perlis, Tianxi Cai.

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
