## [Decision Letter · Decision Letter 0]

5 Dec 2019

PONE-D-19-19412

Integrating questionnaire measures for transdiagnostic psychiatric phenotyping using word2vec

PLOS ONE

Dear Dr. Perlis,

Thank you for submitting your manuscript to PLOS ONE. After careful consideration, we feel that it has merit but does not fully meet PLOS ONE’s publication criteria as it currently stands. Therefore, we invite you to submit a revised version of the manuscript that addresses the points raised during the review process.

We would appreciate receiving your revised manuscript by Jan 19 2020 11:59PM. To enhance the reproducibility of your results, we recommend that if applicable you deposit your laboratory protocols in protocols.io, where a protocol can be assigned its own identifier (DOI) such that it can be cited independently in the future. For instructions see: http://journals.plos.org/plosone/s/submission-guidelines#loc-laboratory-protocols

We look forward to receiving your revised manuscript.

Kind regards,

Sinan Guloksuz, M.D., Ph.D.

Academic Editor

PLOS ONE

Journal Requirements:

1. Thank you for including your competing interests statement; "Dr. Perlis has served on advisory boards or provided consulting to Genomind, RID Ventures, and Takeda, and holds equity in Psy Therapeutics and Outermost Therapeutics. Dr. Perlis is an Associate Editor at JAMA Network Open. The other authors report no conflict of interest."

2.  Please amend the manuscript submission data (via Edit Submission) to include authors Aarón Sonabend Worthalter MA , Amelia M. Pellegrini BA , Stephanie Chan PhD , Hannah E. Brown MD, James N. Rosenquist MD, Tianxi Cai ScD

Reviewers' comments:

Reviewer's Responses to Questions

**Comments to the Author**

1. Is the manuscript technically sound, and do the data support the conclusions?

Reviewer #1: No

Reviewer #2: Yes

2. Has the statistical analysis been performed appropriately and rigorously? 

Reviewer #1: I Don't Know

Reviewer #2: Yes

3. Have the authors made all data underlying the findings in their manuscript fully available?

Reviewer #1: Yes

Reviewer #2: Yes

4. Is the manuscript presented in an intelligible fashion and written in standard English?

Reviewer #1: Yes

Reviewer #2: Yes

5. Review Comments to the Author

Reviewer #1: The authors combine data from multiple clinical rating scales from 310 individuals with and without psychiatric diagnoses and apply word embeddings to transform this data into a larger dimensional representation of the scales, followed by several data reduction steps and techniques, including PCA, CCA and cluster analysis and then finally use these reduced representations in multiple models to distinguish control subjects (divided into super controls and controls) and psychiatric cases and demonstrate rank correlations with cognitive measures.

The major limitation in this paper is that the stated aim, to “validate the clinical utility of the embedding scores,” is never directly tested and instead there are just multiple transformations of scores into high and then low dimensional spaces, resulting ultimately in models which can distinguish psychiatric patients from healthy controls using nine assessments of various clinical symptoms. One key aspect lacking from these multiple tests and manipulations is no direct test of added benefit from applying word embeddings to the scores. Could one not simply take the multidimensional data space of the scales themselves and subject this data space to dimensional reduction? If anything is gained by the additional step of training the word embedding vectors, this is unclear from the paper as written. It is also unclear exactly how transforming a 42-item questionnaire (like the DASS) into a 500-dimensional vector provides additional information, and it appears as though every questionnaire was subjected to this same high dimensional representation. We are given no examples of how the word embedding representations map onto various RDoC domains, DSM diagnoses, or symptoms.

In the discussion, there are statements that this word embedding representation “captures each question’s meaning as opposed to just the ordinal response,” however, this quality of the analysis is not demonstrated at all. Furthermore, since the questions are uniform between participants, how does this technique “quantify how the language is perceived by the responder” when every single participant will answer with an ordinal response.

In the introduction and discussion, there is mention of the five RDoC domains, but there is no explicit mention or investigation of any of the representations in terms of these domains, although I’m sure some could be implied.

The figures are very low resolution and mostly illegible.

Throughout the paper there is mention that these representations were compared to Axis 1 diagnoses, but the actual analyses seem to reduce this to super controls having no lifetime history of Axis 1 Dx, controls having one lifetime Axis 1 that is not MDD, bipolar, schizophrenia or schizoaffective and “cases” having current diagnoses of MDD, bipolar, schizophrenia or schizoaffective. This appears to provide limited clinical utility.

The paper cannot be accepted as written.

Reviewer #2: I have reviewed the article entitled with "Integrating questionnaire measures for transdiagnostic psychiatric phenotyping using word2vec". First of all, I would like to congratulate the authors for their effort to establish a transdiagnostic tool. Hypothesis is well defined, the method is mind-provoking, results of the study might be an important step for the development of future artificial intelligence technology that would allow to screen general psychiatric symptoms apart from the descriptive categorical diagnosis. I would like to clarify a few minor points that are listed below.

1. Authors report that symptom dimensions were chosen considering the RDOC. However, it is difficult to conclude that RDOC describes all prominent dimensions of general psychiatric pathology. Besides, it is still concurrent with the most recent verison of classical descriptive diagnostic categories. Therefore, authors should indicate other evidence for the reason of RDOC choice.

2. It is interesting to see that cognition was chosen as an another domain for this proposed diagnostic tool and CANTAB was used for the evaluation. That would be better to read the reasons in details.

3. Impact of the treatment and also disease related factors on this instrument should be discussed in different aspects for each disorders considering different disease and treatment characteristics.

4. A couple sentences should be added to conclusion in terms of future predictions and contributions to the field

6. PLOS authors have the option to publish the peer review history of their article (what does this mean?). If published, this will include your full peer review and any attached files.

Reviewer #1: No

Reviewer #2: Yes: Kursat Altinbas

---

## [Author Response · Author response to Decision Letter 0]

28 Jan 2020

File attached in submission under Response to Reviewers.

---

## [Decision Letter · Decision Letter 1]

6 Mar 2020

Integrating questionnaire measures for transdiagnostic psychiatric phenotyping using word2vec

PONE-D-19-19412R1

Dear Dr. Perlis,

We are pleased to inform you that your manuscript has been judged scientifically suitable for publication and will be formally accepted for publication once it complies with all outstanding technical requirements.

With kind regards,

Sinan Guloksuz, M.D., Ph.D.

Academic Editor

PLOS ONE

Additional Editor Comments (optional):

Reviewers' comments:

Reviewer's Responses to Questions

**Comments to the Author**

1. If the authors have adequately addressed your comments raised in a previous round of review and you feel that this manuscript is now acceptable for publication, you may indicate that here to bypass the “Comments to the Author” section, enter your conflict of interest statement in the “Confidential to Editor” section, and submit your "Accept" recommendation.

Reviewer #2: All comments have been addressed

2. Is the manuscript technically sound, and do the data support the conclusions?

Reviewer #2: Yes

3. Has the statistical analysis been performed appropriately and rigorously? 

Reviewer #2: Yes

4. Have the authors made all data underlying the findings in their manuscript fully available?

Reviewer #2: Yes

5. Is the manuscript presented in an intelligible fashion and written in standard English?

Reviewer #2: Yes

6. Review Comments to the Author

Reviewer #2: Authors responded all of my previous comments and this revised version of the article is much more acceptable for publication.

7. PLOS authors have the option to publish the peer review history of their article (what does this mean?). If published, this will include your full peer review and any attached files.

Reviewer #2: Yes: Kursat Altinbas

---

## [Editor Report · Acceptance letter]

20 Mar 2020

PONE-D-19-19412R1 

Integrating questionnaire measures for transdiagnostic psychiatric phenotyping using *word2vec*

Dear Dr. Perlis:

I am pleased to inform you that your manuscript has been deemed suitable for publication in PLOS ONE. Congratulations! Your manuscript is now with our production department. 

With kind regards,

on behalf of

Dr. Sinan Guloksuz 

Academic Editor

PLOS ONE